# The Phagosome–Lysosome Fusion Is the Target of a Purified *Quillaja saponin* Extract (PQSE) in Reducing Infection of Fish Macrophages by the Bacterial Pathogen *Piscirickettsia salmonis*

**DOI:** 10.3390/antibiotics10070847

**Published:** 2021-07-12

**Authors:** Hernán D. Cortés, Fernando A. Gómez, Sergio H. Marshall

**Affiliations:** Laboratorio de Genética e Inmunología Molecular, Instituto de Biología, Pontificia Universidad Católica de Valparaíso, Valparaíso 2340000, Chile; fernando.gomez@pucv.cl (F.A.G.); sergio.marshall@pucv.cl (S.H.M.)

**Keywords:** *Quillaja saponaria*, saponin, bacterial, *Piscirickettsia salmonis*, fish macrophage, phagosome–lysosome fusion

## Abstract

*Piscirickettsia salmonis*, the etiological agent of Piscirickettsiosis, is a Gram-negative and facultative intracellular pathogen that has affected the Chilean salmon industry since 1989. The bacterium is highly aggressive and can survive and replicate within fish macrophages using the Dot/Icm secretion system to evade the host’s immune response and spread systemically. To date, no efficient control measures have been developed for this disease; therefore, the producers use large amounts of antibiotics to control this pathogen. In this frame, this work has focused on evaluating the use of saponins from *Quillaja saponaria* as a new alternative to control the Piscirickettsiosis. It has been previously reported that purified extract of *Q. saponaria* (PQSE) displays both antimicrobial activity against pathogenic bacteria and viruses and adjuvant properties. Our results show that PQSE does not present antimicrobial activity against *P. salmonis*, although it reduces *P. salmonis* infection in an in vitro model, promoting the phagosome–lysosome fusion. Additionally, we demonstrate that PQSE modulates the expression of *IL-12* and *IL-10* in infected cells, promoting the immune response against the pathogen and reducing the expression of pathogen virulence genes. These results together strongly argue for specific anti-invasion and anti-intracellular replication effects induced by the PQSE in macrophages.

## 1. Introduction

Piscirickettsiosis (SRS), the main bacterial infectious disease and the main cause of antimicrobial use in the Chilean salmon industry, does not currently have an adequate control measure that can be implemented industry-wide [1,2]. *P. salmonis*, the etiological agent of SRS, has been identified in salmon net-pens in Norway, Canada, Ireland, and Scotland, but with a reduced virulence compared with the Chilean strains [3]. Although public and private efforts have been made in Chile to counteract Piscirickettsiosis (SRS), the lack of efficient strategies has led to the use of significant amounts of antimicrobials, for which there is a global safety concern [4,5,6,7,8,9,10,11].

The effort to try to reduce the use of antimicrobials for Piscirickettsiosis (SRS) control at salmon farms has been based on live/inactivated vaccines [3,12,13], bacteriophages (viruses as bactericidal agents) [14,15], functional diets to reinforce the immune system (essential oils, organic acids, probiotics, prebiotics, and plant extracts) [5,16], and genetic selection for disease resistance [17,18].

The intracellular nature and the complex infection strategies of *P. salmonis* explain in part why an efficient solution to prevent and control the Piscirickettsiosis has yet to be developed, and why new alternatives that overcome this limit are still needed in SRS control [19,20,21,22]. One of the main strategies that *Piscirickettsia salmonis* uses to generate its systemic infection is to infect, survive, and reproduce within cells of the innate immune system with phagocytic capacity, such as macrophages, monocytes, and polymorphonuclear leukocytes [22,23], without generating a cytopathic effect [20,23]. Once *P. salmonis* comes into contact with the host, it is phagocytosed by mucous macrophages in gills, skin, and the gastrointestinal tract, where it manages to establish a niche in the phagosome [22,23]. 

Once the bacteria enter the cells, they are located within phagosomes, vesicles formed by invagination of the plasma membrane during phagocytosis of microbes, which undergo fusion–fission processes with other endocytic organelles, such as lysosomes, to finally form the complex structure called the ‘phago-lysosome’ (P-L) [24]. The phagosome–lysosome fusion, under acidic conditions, degrades the invading pathogen and prevents it from succeeding in its infection [25,26]. Faced with this reality, one of the main cellular responses that is thwarted by intracellular pathogens through virulence mechanisms is phagosome maturation, which in non-infection conditions, or when the pathogen fails to block its maturation, progresses until it joins the lysosome [22,25,26].

Intracellular bacterial pathogens use different strategies that allow them to adhere to, invade, and replicate in host cells and modulate intracellular processes such as membrane trafficking, signaling pathways, metabolism, and cell death and survival [27,28,29]. One of the strategies used by this bacterium is secretion systems (SSs), complex multi-protein transmembrane nanomachines classified as Type I–IX SSs, that form a channel that allows for the exportation of different molecules, including virulence effectors and DNA [29,30]. Within the secretion systems, the Type IV secretion systems (T4SSs) have been described as one of the most adaptable and capable of transferring proteins and nucleic acids to both prokaryotic and eukaryotic cells [30,31]. In *L. pneumophila*, the Dot/Icm nanomachine shapes a secretion channel made of an inner membrane (IM)-anchored DotO ATPase complex. On the other hand, the DotB is an active ATPase recruited from the cytosol to interact with the DotO, forming a complex necessary for the translocation of substrates by the Dot/Icm machine [32].The Dot/Icm T4-BSS may secrete and transfer different bacterial effector proteins into the vacuolar membrane and cytosol of the host cell [33,34].

In *P. salmonis*, some bacterial pathogenicity mechanisms have been described, such as the Type IV-B secretion system, outer membrane vesicles (OMVs) containing different bacterial components (lipopolysaccharides (LPSs), proteins, DNA, and RNA) [3], and the Type IV pili (T4P), a bacterial surface filamentous structure important for adherence to host cell surfaces [35]. Among the above-mentioned pathogenicity mechanisms, the Type IV-B secretion system or Deficient in Organelle Trafficking/Intracellular Multiplication (Dot/Icm) SS has been reported to be the most virulent mechanism in phylogenetically related pathogens, such as *L. pneumophila* and *C. burnetii* [31,36,37,38].

Some genes related to the Dot/Icm T4-BSS found in *L. pneumophila* have been reported in the genome of the *P. salmonis* LF-89 strain, and its putative pathogenic role in cell culture challenges with *P. salmonis* has been determined [39,40]. Gómez et al. [39] demonstrated that four *P. salmonis dot*/*icm* homologue genes (*dotB*, *dotA*, *icmK*, and *icmE*) are expressed either during in vitro tissue culture cell infection or in cell-free media, suggesting their putative constitutive expression. This strategy inhibited phagosome–lysosome fusion, which supports the intracellular survival and multiplication of *P. salmonis* [39]. The Dot/Icm T4-BSS gene expression is upregulated during the different stages of the infection process of *P. salmonis*, especially during the early phases post infection, indicating its important role in the pathogeny of *P. salmonis* [38,39]. A downregulation of these pathogenic genes could affect the bacterial infection process.

Using the Dot/Icm T4-BSS, intracellular bacteria, such as *P. salmonis*, imbalance the innate immune response, changing the early inflammatory response by affecting the interleukin *IL-12* (inflammatory)/*IL-10* (anti-inflammatory) balance (ratio > 1) [41,42,43,44,45,46,47,48]. This change is fundamental to the infection, as it modulates, among other things, the vesicle trafficking inside macrophages and prevents the maturation of the phagosome and its subsequent fusion with lysosomes (non-phagosome–lysosome fusion). It is well established that interleukin 12 (*IL-12*) is critical to counteracting intracellular pathogen infections, while *IL-10* favors the bacteria’s intracellular survival. The *P. Salmonis* LF-89 and EM-90 isolates induce a significant imbalance between *IL-10* (overexpression) and *IL-12* (under-expression) [20,49], which reduces the lytic properties of macrophages. Regarding bacterial characteristic markers, *Piscirickettsia salmonis* has a highly immunogenic heat shock protein (HSP), called “*ChaPs*”, that was isolated from salmonid fish naturally infected with the bacterium and can be used as an indicator of intracellular multiplication of *P. salmonis* [50]. Based on this evidence, this study included the evaluation of the gene expression of *chaPs* in the infection assays with the aim of supporting the monitoring of the intracellular growth of the bacterium.

*Quillaja saponaria* Mol (Quillay) is a native tree from Chile whose extracts have been used to obtain natural detergents called saponins [51]. *Quillaja* extracts contain triterpene saponins, consisting of quillaic acid glycosides, some sugars (glucose, galactose, arabinose, xylose, and rhamnose), along with polyphenols and other minor components [52,53,54]. Saponins have biochemical properties that grant them tenso-active, emulsifier, surfactant, antibacterial, insecticide, antioxidant, and immune-modulating characteristics, so they are widely used in applications for animal feed and health, human food, agriculture, and the cosmetic and pharmaceutical industries [53,55,56,57].

Although there are few publications, *Quillaja* extracts have been reported to have an effect on bacterial invasions [58,59] and modulation of the immune response [55]. However, those few studies did not explore the mode of action (MOA) at the intracellular molecular level, and there are no studies on *Piscirickettsia salmonis*. The purified *Quillaja saponin* extracts (PQSEs) contain a very high concentration of saponins, between 65 and 96% *w*/*w*, unlike non-purified *Quillaja saponin*, which contains other components, such as polysaccharides, polyphenols, and oligosaccharides [60,61]. With the above background, we question whether PQSEs affect the infection process of *Piscirickettsia salmonis* in an Atlantic salmon macrophage cell line (SHK1), and whether this effect could be mediated by a modulation of the inflammatory and anti-inflammatory balance and favor the phagosome–lysosome fusion.

In general, we found that purified *Quillaja saponaria* extracts (PQSEs) do not have a direct antimicrobial effect on *P. salmonis* (MIC > 30 mg/mL) but, if they reduce the invasion and intracellular replication of *P. salmonis* in macrophages (SHK-1 fish cell lines), downregulate bacterial genes encoding for virulence factors such as *dotB* ATPase from the T4-BSS and the chaperone protein *chaPs* (a shock thermic protein), upregulate genes encoding for proinflammatory *IL-12* and anti-inflammatory *IL-10* cytokines, favoring *IL-12* over *IL-10*, characteristic of an activated-destructive macrophage, and favor the phagosome–lysosome fusion.

In summary, we found that purified Quillaja saponin extract (PQSE) reduces the infectious process of *P. salmonis* in vitro and, at the same time, promotes a suitable immune marker (*IL-12*/*IL-10*) balance and the phagosome–lysosome fusion, which are critical factors for counteracting a *P. salmonis* infection.

## 2. Materials and Methods

The purified *Quillaja saponaria* extract used in this study was supplied by Desert King (Chile) and corresponds to an aqueous extract obtained from the biomass of the Chilean *Quillaja saponaria* tree (Quillay) [51,62]. PQSE is a highly purified *Quillaja saponaria* extract in powder form that mainly contains triterpenic saponins (96% saponins, *w*/*w*) and around 4% other components in minor quantities, named the non-saponin fraction (NSF, Figure 1). The NSF in the *Quillaja saponin* extracts includes a blend of phenolic, polysaccharide, and oligosaccharide components [60].

### 2.1. Bacterial and Cell Culture

The *P. salmonis* strain used was LF-89 (ATCC VR-1361), which was grown at 18 °C on blood agar plates (5 g/L peptone, 5 g/L yeast extract, 15 g/L tryptone, 10 g/L glucose, 12 g/L agar, and 5% lamb blood) supplemented with 0.1% L-Cysteine [63]. For the liquid culture, a single colony of *P. salmonis* was taken, inoculated in 5 mL of BM3 liquid, and incubated at 18°C and 100 rpm of agitation [64].

The cells used in this study correspond to the SHK-1 macrophage cell line derived from *Salmo salar* Head Kidney-1. These cells were grown at 20 °C with Leibovitz’s L-15 medium supplemented with 10% Fetal Bovine Serum (FBS) (Gibco), 4 mM L-Glutamine, 50 µg/mL Gentamicin, and 40 µM 2-mercaptoethanol [65].

### 2.2. Direct Quillaja Extract Antimicrobial Analysis

The ability of the *Quillaja* extracts to directly inactivate *P. salmonis* was evaluated by means of a standard microplate dilution test in the complex culture medium BM3 [64], determining the Minimum Inhibitory Concentration (MIC). Briefly, a *P. salmonis* inoculum was grown in the BM3 medium overnight at 18 °C with shaking at 100 rpm. Bacterial growth was determined by turbidimetry at the OD_600_ absorbance on a spectrophotometer, making growth measurements every 4 h [66]. The methodology used to determine the MIC was an adaptation of the protocol defined in the European Committee for Antimicrobial Susceptibility Testing, with 15 h of exposure with the products at 18 °C on the bacteria in the BM3 culture medium [67].

A stock solution from the PQSE (96% saponins, *w*/*w*) was prepared in 5.0% BM3 medium (0.5 g/10 mL). In a 96-well microplate, two base dilutions of these PQSEs were made in BM3 medium, starting at a concentration of 1 µg/mL and going up to 32,000 µg/mL (32 K). *P. salmonis* was inoculated at each dilution of the plate at an OD_600_ between 0.3 and 0.4. These dilutions were distributed on the microplate, including a target (B) without bacteria and a positive control (0) with bacteria (BM3 medium + *P salmonis*). Each assay was performed in triplicate.

### 2.3. Cytotoxicity

The cytotoxicity of the Quillaja extract (PQSE) was evaluated on the SHK-1 cell line, for which the cells were placed in 96-well culture microplates, using the same conditions described above, until 90% confluence was reached. To evaluate the cytotoxicity of the PQSE on SHK-1 cells, a colorimetric assay was carried out using the compound MTS (3-(4,5-dimethylthiazol-2-yl)-5-(3-carboxymethoxyphenyl)-2-(4-sulfophenyl)-2H-tetrazolium, inner salt). This method quantifies by absorbance values the cytotoxic effect that a molecule may have on cell viability [68]. The evaluated PQSE doses were 0.4, 0.8, 1.6, 3.1, 6.2, 12.5, 25, 50, and 100 µg/mL for each of the PQSEs, which were incubated for 24 h.

### 2.4. Internalization of P. salmonis in SHK-1

A culture of the SHK-1 cell line was grown in L-15 medium supplemented with 10% FBS for 3 to 4 days until 100% confluence was reached. Then, 5 × 10^5^ cells per well were seeded in a 24-well plate and incubated for 24 h. A dose of 1 µg/mL of PQSE was selected, based on the previous cytotoxicity test in MTS, in such a way that it did not affect the cell viability. *P. salmonis* LF-89 was grown overnight in 3 mL of BM3 broth under two different conditions: (1) BM3 alone; and (2) BM3 + PQSE, 1 µg/mL.

After incubation with the PQSE, the medium was removed from the cells and 5 washes with PBS were performed. The cells were subsequently infected with *P. salmonis* at an OD of 0.3–0.4 (MOI: 10) [23] and incubated for 0.5 h (for early internalization) and 3 h (for late internalization) at 18 °C and without agitation. At 0.5 h and 3 h post infection, respectively, the medium was removed from the cells, washed three times with PBS, and fixed with methanol for 15 min at 4 °C.

To observe the bacteria that entered the cells, an immunofluorescence microscope was used, previously staining *P. salmonis* and the cell membranes. To detect *P. salmonis* in infected cells, indirect immunofluorescence was performed, using the commercial kit “SRS-Fluoro Test”, which uses a mixture of monoclonal antibodies specific to *P. salmonis* (SRS oligoclonal) that react with specific antigens of the bacterium. After washing to remove the free antibodies, a FITC-conjugated anti-mouse IgG antibody was added, which allowed us to determine the presence of *P. salmonis*. Regarding the membranes, these were stained with Octadecyl-rhodamine B (R18), a probe that is incorporated into the lipids of unmarked membranes, resulting in dilution of the probe and fluorescence emissions.

### 2.5. Proliferation of P. salmonis in SHK-1 Cells Pre-Incubated with PQSE

Four activities were proposed to verify whether the bacteria that enter cells pre-incubated with PQSE remain viable, whether they are proliferating, and whether the bacteria alter some cellular mechanism within the macrophage (SHK-1). These activities were: (A) determine the bacterial load (bacterial DNA) using ITS by qRT-PCR; (B) evaluate the transcriptional activity of the *dotB* gene from the Type IV B Secretion System (T4-BSS), and the *P. salmonis chaPs* protein in SHK-1 cells pre-incubated with PQSE and challenged with the bacteria, using qRT-PCR; (C) evaluate the expression of *IL-12* and *IL-10* cytokines in SHK-1 cells pre-incubated with PQSE and challenged with *P. salmonis* using qRT-PCR; and (D) evaluate the formation of “phago-lysosomes” in SHK-1 cells pre-incubated with PQSE and challenged with *P. salmonis*. For the intracellular proliferation test of *P. salmonis* in SHK-1 cells pre-incubated with PQSE, infection kinetics were determined using a protocol similar to the internalization test, without including the immunofluorescence phase and taking measurements at 1 hpi (to validate the early internalization) and at 72 hpi (to evaluate the bacterial proliferation). 

### 2.6. Quantification of P. salmonis DNA: Bacterial Load

Bacterial load was determined by absolute quantification of *P. salmonis* DNA by qRT-PCR of the internal transcribed spacer (ITS), as was previously described [69]. For amplification of the ITS, RTS1 (5′-TGA TTT TAT TGT TTA GTG AGA ATG a-3′) and RTS4 (5′-ATG CAC TTA TTC ACT TGA TCA A-3′) primers previously standardized and reported [70] were used. The genomic DNA was extracted following the Genomic DNA Purification Protocol of Thermo Scientific’s GeneJET Genomic DNA Purification Kit #K0721, #K0722. The thermocycler protocol was calibrated with the following cyclical profile: 95 °C for 3 min for the initial denaturation, and 40 cycles of 95 °C for 15 s, 51 °C for 15 s, and 60 °C for 20 s. For all qPCR amplifications, the ThermoFisher PowerUp SYBR Green Master Mix was used.

To develop the *P. salmonis* qPCR standards, the ITS region was amplified based on a public standardized procedure [69]. Briefly, the procedure considered the cloning of the ITS amplicon into the pCR 2.1 vector (Invitrogen Inc., Waltham, MA, USA). For the isolation of plasmid DNA, a Miniprep Kit (Quiagen, Hilden, Germany) was used, following the recommendations of the manufacturer. After that, the purified ITS plasmid was quantified, and diluted series were prepared in DNase–RNase-free water to obtain the end concentrations from 1 to 1010 copies of genome equivalents. Finally, 2 µL from each dilution was taken in triplicate for real-time PCR and to develop the standard curve used to quantify the *P. salmonis* DNA [69]. 

A summary with the sequence details of RTS1, RTS4, and all primers used in this study is shown in Table 1.

### 2.7. Transcription of the dotB, chaPs, IL-10, and IL-12 Genes

To quantify the transcription of the virulence genes *dotB* and *chaPs*, as well as the interleukins *IL-10* and *IL-12*, after each infection time (1 hpi and 72 hpi), cells were harvested for RNA extraction. At each point of the infection kinetics, cells were harvested from the cell culture plates with a cell scraper and centrifuged in the same medium at 500 rpm for 10 min at 4 °C. The resulting pellet was processed for RNA extraction with Trizol^®^ (Invitrogen, Waltham, MA, USA) according to the manufacturer’s instructions. RNA concentrations were measured on a Nanodrop-1000 spectrophotometer and kept at −80 °C until use. At each infection time, two biological replicates were made.

### 2.8. cDNA Synthesis to Quantify the Transcription of dotB, chaPs, IL-10, and IL-12 Genes

For the cDNA synthesis, 2 µg of the RNA obtained from the infected cell lines was used, which was treated with the enzyme DNase RQ1 (Promega) in a reaction volume of 10 µL for 30 min at 37 °C in order to eliminate possible DNA contamination. After the inactivation of the DNAse at 65 °C for 10 min, the synthesis of the cDNA was performed using the Affinity Script qRT-PCR cDNA synthesis kit (Agilent) and random primers (6-bp primers that hybridize to the different regions of the tempered RNA randomly). The synthesis reaction was performed in a final volume of 30 µL, containing 2 µg of RNA, 1× of cDNA synthesis Master Mix, 3 ng of random primers, and 5U of Affinity Script RT/RNase Block enzyme mixture. The reaction was carried out in a thermocycler under the specific temperature conditions for each gene. Following this, the cDNA was stored at −80 °C until use.

The *dotB* and *chaPs* genes expression analysis were done by relative quantification by q-PCR using the 2^−^^ΔΔCt^ method [71] and taking *sdhA* as the reference normalizing gene (housekeeping gene) [72]. The *sdhA* gene had previously been selected by Flores-Herrera et al. (2018), who evaluated different genes under growth and stress conditions and concluded that *sdhA* was the most reliable housekeeping gene for qPCR expression analyses of *P. salmonis*. To determine the expression of the *dotB* gene, the primers previously reported for the bacterium [39] were used.

Based on previous studies with reference genes [73], Elongation Factor 1-Alpha (ELF1A) was selected to control the host cell to normalize the qPCR and the amount of RNA at each point of kinetics used. The samples were amplified and detected in a Real-Time PCR CFX96 system (Biorad) using the following cycle profile for the *dotB* gene: 95 °C for 3 min for initial denaturation, 95 °C for 15 s, followed by 49 cycles of 58 °C for 15 s and 60 °C for 20 s.

For the amplification of ELF1A (*elf1A*), the primers ELF1A-For and ELF1A-Rev were used, as described in previous studies [73], with a cycle profile of 95 °C for 3 min for initial denaturation and 40 cycles of 95 °C for 15 s, 56 °C for 15 s, and 60 °C for 20 s. All PCRs were evaluated in each biological replica, and each sample was run in triplicate.

For all cases, the q-PCR efficiencies were calculated from the slope according to the equation: E = 10^(−1/slope)^ [74]. The Ct (threshold cycle) values of the CFX Manager software (Biorad) were transformed to relative amounts as previously described [73]. For the Ct conversion to relative amounts, the reaction efficiencies were used. The relative expressions of the *dotB* gene and the *chaPs* protein from the bacteria were calculated using the values obtained for *sdhA* (the normalizing gene) from each evaluation.

### 2.9. Expression of IL-12 and IL-10 Genes by qRT-PCR in SHK-1 Cells Pre-Incubated with PQSE

The evaluation of the PQSE’s effect on the gene expression of interleukins *IL-10* and *IL-12* at the cellular level was determined by q-PCR using the relative quantification of the Livak method [71]. Briefly, infection kinetics were performed in SHK-1 cells and samples were taken at 1, 24, and 72 h post infection (hpi). Cells were plated in 12-well plates to 90% confluence. Total RNA was purified using Trizol LS R reagent (Invitrogen) according to the manufacturer’s instructions. The cDNA was synthesized from 2 µg of RNA pre-treated with DNase (Promega, Madison, WI, USA) using an enzyme M-MLV reverse transcriptase (Promega, Madison, WI, USA) in the presence of oligo (dt) according to the manufacturer’s instructions.

### 2.10. Evaluation of Phagosome–Lysosome (P-L) Fusion

To determine the ability of *P. salmonis* to prevent phagosome–lysosome fusion after phagocytosis, immunofluorescence staining was performed on the SHK-1 cell line.

The study consisted of three phases. Phase I comprised the growth of a SHK-1 cell culture to 80% confluence. As a control treatment, 1 mL of an overnight culture of *P. salmonis* was inactivated with 3% formaldehyde (F) for 24 h at 4 °C. After the inactivation time, this 1 mL of inactivated *P. salmonis* (inactivated positive control, IPs) was washed 3 times in 1× PBS and re-suspended in sterile 1× PBS. Finally, it was used to infect cells and incubate them for 24 h.

To evaluate the effect of the addition of *Quillaja* Extract (PQSE), three conditions were considered: (A) pre-treatment of SHK-1 cells for 4 h before the challenge with *P. salmonis* for 3 h; (B) co-treatment of SHK-1 cells with PQSE and simultaneous challenge with *P. salmonis* for 4 h; and (C) post-treatment of SHK-1 cells with PQSE (4 h) after challenge with *P. salmonis* for 3 h. The cells were infected with 67 µL of either a virulent (positive control, vPs) or formalin-inactivated *P. salmonis* (iPs) inoculum at a MOI of 10. The overnight culture of *P. salmonis* was used when it reached an optical density (OD) of 0.3 to 0.4, equivalent to an approximate value of 1.5 × 10^7^ bacteria/mL.

For lysosome labeling, cells infected with live (vPs) and inactivated (iPs) *P. salmonis* were incubated at 20 °C for 1 h in the dark, with 75 nM (3 µL/poc) of the LysoTracker Red DND-99 kit (Thermo Fisher, Waltham, MA, USA) diluted in L-15 medium. It was subsequently washed 3 times with sterile 1× PBS. 

The LysoTracker^®^ Red DND-99 probe (Thermo Fisher, Waltham, MA, USA) is a fluorescent red dye that, due to its high selectivity for acidic organelles, very specifically allows for the staining and monitoring of organelles such as lysosomes. LysoTracker^®^ Red probes are composed of a partially protonated and weak base-bound fluorophore with neutral pH (excitation/emission: 577/590 nm), a condition that allows them to easily cross cell membranes and label living cells (Thermo Fisher, Waltham, MA, USA).

Phase II was “immunofluorescence”. This included steps that are generally used in fixed cell immunofluorescence protocols: fixation, permeabilization, incubation with a primary antibody, incubation with a secondary antibody, and mounting.

In the fixation step, cells were fixed for 15–30 min in 4% paraformaldehyde (PFA) diluted in PBS 1× at pH 7. The stock of 30% PFA was brought to 4% in PBS1 1×, and heated to less than 70 °C (40 °C) × 10 min in a shaker, if turbid, until clear. It was performed inside the hood and the tube (50 mL) and the PFA was covered with parafilm for protection against gas volatilization. Then, the pH was measured and stabilized at 7.3. The 4% PFA dilution was filtered and stored at 4 °C before use. After the fixation of cells (600 µL/well), it was washed with PBS 1× 3–5 times. Then, PBS was added again and left for about 18 h at 4 °C.

The cell membrane permeabilization stage was performed to facilitate the entry of the antibodies. For this, 0.1% Triton X-100 detergent (Sigma-Aldrich, Taufkirchen, Germany) was used. The fixed cells were incubated for 20 min and then washed 3 times with PBS 1×. 

In the incubation phase with antibodies, the labeling of *P. salmonis* was carried out by indirect immunofluorescence using the SRS-Fluorotest kit (Ango). For this, the primary antibody (Ab1°) was diluted to 1/100 *v*/*v* in the dilution solution of the Fluorotest kit (total: 50 µL of Ab1° in 5 mL of the dilution solution). Then, 500 µL/well of diluted Ab100 at 1/100 was added, and the cells were incubated at room temperature for 30 min with the diluted oligoclonal Ab1. Finally, it was washed with PBS. For incubation with the secondary antibody (Ab 2°), the cells were incubated in the dark for 30 min, with the Ab2° ligated to Alexa Fluor 647 goat anti-mouse IgG. The Ab2° was diluted to 1/200. Subsequently, the cells were washed with PBS. For nucleus staining, TOPRO-3 fluorescent dye was used for a maximum of 5 min. The TOPRO 3 was diluted to 1:2000. 

For the mounting phase, cells previously washed in PBS were mounted with Dako mounting medium (Invitrogen). The samples were analyzed using a Leica TCS SP5II confocal spectral microscope (Leica Mycrosystems Inc., Wetzlar, Germany). Images were obtained with a 40×/1.25 objective HCX PL APO CS Oil (Leica Microsystems Inc., Wetzlar, Germany).

Figure 2 shows the conditions used to determine whether the pre-treatment, co-treatment, and post-treatment with *Quillaja* extract (PQSE) altered the phagosome–lysosome fusion in a kinetics of infection with *P. salmonis* in SHK-1 cells. Each plate in the assay contained approximately 100,000 cells, and the Multiplicity of Infection (MOI) (bacteria/cell) was 10 in accordance with previous reference studies [23].

### 2.11. Statistical Analysis

The results correspond to an average of two experiments (n = 2) and three replicates for each concentration (3 rep/concentration). The results of each experiment are expressed as the mean ± the standard deviation. For the statistical analysis that allowed us to determine significant differences between the experimental and biological replicates of the different groups (at a significance level of *p* < 0.05), the non-parametric Mann–Whitney test was used for the results of the gene expression of the *dotB* gene, the *ChaPs* protein, and the interleukins *IL-10* and *IL-12*. To verify the results, an additional analysis of relative quantification using the 2^−ΔΔCt^ method [71] was performed. To evaluate the time and treatment effects on the internalization of *P. salmonis* in SHK-1 cells, a Manova analysis was used.

## 3. Results

### 3.1. SHK-1 Cell Lines Are Sensitive to the Quillaja Extract Levels

Using a MTS mitochondrial functionality test, the viability of SHK-1 cells pre-incubated with *Quillaja saponin* extract (PQSE) was determined.

As indicated in Figure 3, the viability of SHK-1 cells was higher than 90% with doses until 6.25 µg/mL of PQSE. However, at 12.5 µg/mL of PQSE (equivalent to 12.0 µg/mL of saponin), the cell viability was reduced to around 60%.

### 3.2. The Purified Quillaja Saponin Extracts (PQSEs) Are Not Toxic to P. salmonis at Physiological Doses for the Cell

In order to determine whether a purified *Quillaja saponaria* extract (96% triterpenoid saponins) could have a direct effect on the *P. salmonis* viability, a Minimum Inhibitory Concentration (MIC) test of the PQSE in BM3, a complex cell-free culture medium specific to the bacteria, was performed. The minimum inhibitory concentration (MIC) of the PQSE is the lowest concentration of the extract that can inhibit the growth of *P. salmonis*.

As a result, it was found that remarkably high doses of PQSE are required to inhibit the growth of the bacteria. Although PQSE doses greater than 1 mg/mL progressively reduced the growth of *P. salmonis*, including doses of 30 mg/mL, they did not fully inhibit its growth (MIC > 30 mg/mL). Based on this, we can say that, compared with Florfenicol, which is the antibiotic most widely used to control *P. salmonis*, the PQSE does not have antimicrobial effects against this bacterium at physiological doses for the cell (non-toxic doses).

### 3.3. Quillaja Saponaria Extracts Reduce the Internalization of P. salmonis in SHK-1

As shown in Figure 4, SHK-1 cells treated for 4 h with 0.5 µg/mL of PQSE, prior to infection with *P. salmonis*, significantly reduced (*p* < 0.05) the internalization of the bacterium in both evaluation times (0.5 and 3 hpi) with respect to the positive control (*P. salmonis*).

The *P. salmonis* internalization test showed that SHK-1 cells pre-treated for 4 h with 0.5 µg/mL of PQSE significantly reduced (*p* < 0.05) the entry of *P. salmonis* into the cells at both 0.5 and 3 hpi. In experiments not included in this text, we found that doses of 1.0 and 2.0 µg/mL of PQSE also reduced the internalization of *P. salmonis* into the cells; however, the lowest dose with a significant effect was 0.5 µg/mL, so this dose was selected for subsequent evaluations.

### 3.4. SHK-1 Cells Pre-Incubated with PQSE Reduce the Intracellular Proliferation (72 hpi) of P. salmonis

As shown in Table 2 (Panel B), SHK-1 cells pre-incubated for four (4) hours with PQSE reduced the intracellular proliferation of *P. salmonis* as measured by absolute quantification of *P. salmonis* DNA at 1 hpi and 72 hpi. Pre-treatment of SHK-1 cells with PQSE inhibited the proliferation of *P. salmonis* by 80% at 1 hpi (Table 2, Panel A) and by 76% at 72 hpi (Table 2, Panel B).

### 3.5. SHK-1 Cells Pre-Treated with PQSE Reduce the Gene Expression of dotB (T4-BSS) and chaPs of P. salmonis

As seen in Figure 5; Figure 6, SHK-1 cells pre-treated for 4 h with a highly purified saponin *Quillaja* extract (PQSE, 96% saponins *w*/*w*), prior to challenge with *P. salmonis*, significantly reduced (*p* < 0.05) the gene expression of the virulence factor *dotB*, which encodes for the ATPase of the *dot*/*Icm* secretion system, and the gene expression of the *chaPs* protein with respect to the positive control (PC) at 24 and 72 hpi.

### 3.6. SHK-1 Cells Pre-Incubated with PQSE Reduce the Gene Expression of Interleukin 10 (IL-10) and Favor the Expression of IL-12 during the Acute Phase of P. salmonis Infection (<72 hpi)

As seen in Figure 7 and Figure 8, SHK-1 cells treated for 4 h with a PQSE, prior to the challenge with *P. salmonis*, significantly reduced (*p* < 0.05) the gene expression of Interleukin 10 (*IL-10*) at 24 and 72 hpi with respect to the positive control (PC, bacteria only), while the *IL-12* expression increased significantly at 24 h compared with the positive control (PC, *P. salmonis* only).

### 3.7. Purified Quillaja Saponaria Extracts (PQSEs) Favor the Formation of Phago-Lysosomes in SHK-1 Cells

This study confirmed that the “live” virulent strains of *P. salmonis* (vPs) have the ability to significantly (*p* < 0.05) reduce the degree of “phagosome conversion” and P-L fusion (only 9% of colocalizations), whereas formaldehyde-inactivated strains of *P. salmonis* (iPs) partially blocked phagosome maturation and P-L fusion, as shown in Figure 9, allowing for the maturation of some phagosomes and their binding to lysosomes (26%).

The phagosome–lysosome (P-L) fusion is a cell response mechanism that, under acidic conditions, degrades invading pathogens and prevents them from proliferating and succeeding in their infection [25,26]. Given this critical cellular function, various pathogens, such as *P. salmonis*, through virulence mechanisms, block the P-L fusion. In non-infection conditions or when the pathogen fails to block its maturation, the process continues and the phagosome–lysosome fusion occurs [25,26].

As shown in Figure 10, Figure 11, Figure 12, Figure 13 and Figure 14, SHK-1 cells treated with PQSE under three different conditions (“pre-incubation”, “co-incubation”, and “post-incubation” with PQSE) and challenged with a “live” virulent strain of *P. salmonis* (vPs, LF89) induced phagosome maturation and the phagosome–lysosome (P-L) fusion (colocalization).

## 4. Discussion

This study shows that purified *Quillaja saponaria* extracts (PQSEs) reduce the internalization and intracellular proliferation of *P. salmonis* in a macrophage fish cell line (SHK-1). The findings show that the incubation of macrophages with PQSE reduced the gene expression of *dotB* (from the T4-BSS) and the *P. salmonis chaPs* protein, generate a modulation of the balance of proinflammatory (*IL-12*) and anti-inflammatory (*IL-10*) interleukins that favors *IL-12* expression over *IL-10* expression, and increases significantly the phagosome–lysosome fusion. Moreover, the MIC determination in this study showed that PQSE does not have an antimicrobial effect at a physiological dose for the cell line.

### 4.1. SHK-1 Cell Lines Were Sensitive to Quillaja saponaria Extracts

PQSEs contain saponins, which have detergent effects when interacting with lipids; so, depending on the dose, they can cause changes in cell membranes [75]. The toxicity of PQSE with different levels of purification in saponins has been explored in the Atlantic Salmon kidney cell line ASK, and cytotoxic effects between 14 and 21 µg/mL in terms of PQSE were reported [58,76]. That result is in line with our results, considering that, in our study, a PQSE with 96% saponins (*w*/*w*), in doses greater than 12.5 µg/mL of product (12.0 µg/mL of saponin), reduced the SHK-1 cell viability to 60%. The toxicity effect of the PQSE on the fish cell line SHK-1 (>12.5 µg/mL) was higher than the dose needed to reduce the internalization and proliferation of *P. salmonis* (0.5 µg/mL).

### 4.2. PQSE Does Not Have a Direct Antimicrobial Effect on P. salmonis at Physiological Doses to the Host Cell

The MIC of PQSE on *P. salmonis*, developed in the BM3-free cell medium, was 30,000 µg/mL, which is in the line with a previous MIC study evaluating other *Quillaja saponaria* extracts on *P. salmonis* in a PSA broth, where a MIC higher than 11,500 μg/mL for *P. salmonis* was reported [76].

In contrast, other researchers showed that *Quillaja* extracts rich in saponins have antibacterial activity against *Salmonella typhimurium*, *Staphylococcus aureus,* and *E. coli*, obtaining a MIC of 100 µg/mL [77]. This study showed that a guar gum flour methanol extract, commercial *Quillaja* saponins, and soy and *Yucca schidigera* saponins have significantly different hemolytic and antibacterial activities against Gram-negative and positive bacteria at the same saponin concentration. When comparing soy and Yucca saponins, soy saponins did not have antibacterial activity against any of the analyzed bacteria (*Staphylococcus aureus*, *Salmonella typhimurium,* and *E. coli*) at the analyzed saponin concentrations, while the Yucca saponin had no antibacterial activity against Gram-negative bacteria at the concentrations analyzed [77].

Other studies also demonstrated the strong antibacterial activity of an aqueous *Quillaja saponaria* extract against four strains of *E. coli* producing the Shiga toxin (STEC) O157:H7 and six non-O157 STEC serotypes [78,79]. When the treatments were done, the four O157:H7 strains had a CFU of 7.5 log and, after 16 h, the counts were reduced to 6.79 and 3.5 log CFU at room temperature. Looking for possible modes of action, analyses were performed using scanning electron microscopy, and damage to the cell membranes of the treated bacteria was found [79,80].

The effects on the growth of bacteria and the MIC of PQSEs can vary widely depending on the EQ used. These extracts have various phytochemical components that, depending on the level and method of extraction, can generate products with a variable composition, leading to variable MIC results [76,81]. According to the results obtained, the PQSEs used in this work do not have a significant effect on the growth of *P. salmonis*; therefore, the mechanism of action of saponins during infection with *P. salmonis* seems to be associated with promoting host cell defense pathways.

### 4.3. PQSEs Reduce the Internalization and Intracellular Proliferation of P. salmonis

Some in vitro and in vivo studies suggest that *Quillaja* saponins can “cover” host cells and prevent the binding of different types of viruses, covered or not, reducing the availability of binding sites for viral pathogens [82,83]. This mechanism of reduction to membrane binding sites has also been suggested to explain the reduction in adherence and internalization of some bacterial strains in HeLa cells treated with saponins [59]. Regarding the effects on macrophage cell lines, our study showed that SHK-1 macrophage-like cells, when incubated with PQSE, reduce the internalization and intracellular proliferation of *P. salmonis*, being phagosome maturation and phagosome–lysosome fusion cellular processes encouraged by PQSE.

The functional analysis of phagosome maturation validated what has been previously demonstrated; i.e., that the pathogenicity of virulent *P. salmonis* (vPs) lies, in part, in its ability to induce a limited lysosomal response in the acute phase of the infection (1 hpi), blocking the phagosome maturation and the “phagosome–lysosome” (P-L) fusion, unlike what happens with an inactivated *P. salmonis* (iPs) strain, in which it is possible to observe a higher degree of P-L fusion than in a virulent one [19,22,39].

In the present study, SHK-1 cells treated with PQSE promoted the trafficking of bacteria to lysosomes, favoring the P-L fusion. It should be noted that this is the first time that a PQSE has been reported to promote an important physiological process at the intracellular level.

Our results show that the reduction of the internalization and intracellular proliferation of virulent strains of *P. salmonis* LF89 (vPs) in SHK-1 cells treated with PQSE occurred at 25 times lower doses (0.5 µg/mL) relative to the dose at which PQSE reduced the cell viability to 60% (12.5 µg/mL). Collectively, the results show that SHK-1 cells treated with PQSE manage to reduce the intracellular proliferation of *P. salmonis* by means of three mechanisms: (1) reducing the expression of the *dotB* (T4-BSS) and *chaPs* protein genes, virulence factors that are expressed in the infective process of *P. salmonis*; (2) modulating the balance of the proinflammatory/anti-inflammatory interleukins *IL-12*/*IL-10*, favoring *IL-12* (proinflammatory) expression over *IL-10* (anti-inflammatory) expression during the first hours post-infection (less than 72 hpi), which favors the control of intracellular pathogens such as *P. salmonis*; and (3) promoting the “phagosome–lysosome” fusion, a fundamental cellular mechanism to combat intracellular pathogens.

A study using Red Ginseng Saponin Fraction-A (RGSF-A) found that its addition to cell lines reduces the adherence, internalization, and intracellular growth of *Brucella abortus* and produces a reduction in the intracellular replication of the bacterium due to the increase in phagosome–lysosome fusion in macrophages [84]. 

*P. salmonis* can reduce the expression of some genes relevant to a specific cell-mediated immunity in the head kidney, such as T lymphocyte receptors, TCR-alpha, TCR-beta, and CD8. This gives to *P. salmonis* the capacity to reduce the activation of T-lymphocytes, which are effector cells against intracellular pathogens [85,86]. The interleukin evaluation in our study showed that *P. salmonis* reverses the balance of *IL-10* and *IL-12*, favoring the expression of *IL-10* and reducing the expression of *IL-12*. This effect induces a "shutdown" of the macrophage in its ability to degrade the pathogen and alters subsequent steps such as the presentation of peptides in MHCII, the formation of the “immunological synapse”, the interface between the antigen-presenting cell and a T/B lymphocyte or Natural Killer cell, the reduction in production of gamma interferon, and the induction of Cytotoxic T Lymphocytes (CD8^+^), which are key in the control of intracellular pathogens. This guarantees the intracellular survival of bacteria.

Reports in mammals have shown that Balb/c mice, supplemented orally with *Quillaja* extracts, can change the profile of cytokines associated with allergies and the antigen-specific immune response [87]. The study found that Quillaja extracts could suppress the ovalbumin-induced, IgE-mediated allergic response by promoting a Th1-dominant immune response.

### 4.4. PQSE Disrupts Cell Membranes, a Mode of Action That Could Help Reduce the Internalization of P. salmonis in SHK-1 Cells

A reduction in the adherence and entry of a virus to PQSE-incubated cell lines was previously reported with reovirus serotype 3 (ST3) and rhesus rotavirus (MMU 18006) in mouse L929 fibroblasts (ATCC CCL-1) and the African green monkey kidney cell line MA-104 (ATCC CRL-2378.1) [82,83]. These studies demonstrated that *Quillaja* extract concentrations as high as 1000 µg/mL were not cytotoxic to those cells, but concentrations as low as 1.0 µg/mL blocked the binding and infection of rotavirus and reovirus. The low dose of PQSE used in that study coincides with the dose (1.0 µg/mL of PQSE) that generated the best protection results against *P. salmonis* infection in our study, although it differed in the susceptibility of the cell lines, as the fish lines were more sensitive [83,88,89].

A study evaluating the potential of *Quillaja Saponaria* and *Yucca schidigera* extracts to reduce the adherence and internalization of bacterial strains demonstrated their affinity for cell membrane cholesterol. This study suggests that pre-treatment of cells for 6 or 24 h with *Quillaja* and *Yucca* extracts may modulate the cholesterol levels in the cell membrane and could explain the reduction observed in the internalization of strains of some food and waterborne bacterial pathogens, such as *Escherichia coli* 0157:H7, *Yersinia enterocolitica*, *Listeria monocytogenes*, two strains of *Salmonella enterica serovar typhimurium*, *Shigella flexneri,* and *Vibrio cholerae* [59].

On the other hand, saponins interact with the lipid domains (lipid rafts) of cell membranes, especially with cell membrane cholesterol, through their aglycone and lipophilic functional groups, such as the acyl group, which are able to generate changes in the fluidity of the membrane [55,59,90,91,92]. Since it can increase membrane fluidity, it is possible that the conformational changes that ATPases undergo during their transport cycle can be facilitated [59,90]. The fluidity of the membranes controls the enzymatic activity of the biological membranes. This plays an important role in ion transport, so the ability of saponins to affect this parameter could explain their impact on the availability of the functional receptors and membrane proteins with which pathogens need to interact. 

The plasma membrane is not homogeneous, and there are micro domains in its composition called lipid rafts that are characterized by being very rich in cholesterol and glycosphingolipids and related to cellular processes such as the classification of membranes and signal transduction [93]. Studies have observed that receptor molecules involved in the cellular activation of lipopolysaccharide (LPS), such as CD14, Toll-like receptor 4 (TLR4), heat shock proteins (HSPs) 70 and 90, chemokine receptor 4 (CXCR4), and growth differentiation factor 5 (GDF5), are present in lipid rafts after stimulation with LPS [94]. The interaction of saponins with membrane cholesterol is known to generate conformational changes [91]. Studies with *Quillaja saponins* have shown that they interact with cholesterol in cell membranes, forming complexes [75]. It has been shown that saponins can reversibly induce the release and/or reduce the uptake of cholesterol from cell membranes [95]. Studies with influenza A viruses show a loss of infectivity of viral particles due to effects mediated by cholesterol depletion at the cell membrane level, causing disruption in the lipid domains, changes in the structural integrity of the viral membrane, filtration of viral proteins, holes in the viral envelope, and alteration of the structure of the virus [96].

On the other hand, PQSE could reduce the internalization and early trafficking of *P. salmonis* in the cell, disturbing Clathrin in lipid rafts.

A study with *Brucella abortus*, an intracellular pathogen that circumvents host defenses, has shown that the association of Clathrin with lipid rafts is essential to host–pathogen interactions during infection [97]. The study found that Clathrin and dynamin were concentrated in lipid rafts during *B*. *Abortus* infection and that *B. Abortus* intracellular entry into and survival in HeLa cells were blocked by Clathrin inhibition. Equally, *P. salmonis* enters the cell via Clathrin-mediated endocytosis [21,98]. The addition of PQSE to SHK-1 cells, due to the ability of saponins to interact with cholesterol, could, to some extent, disorganize rafts that are in connection with Clathrin and reduce the efficiency of bacterial phagocytosis through this route.

Finally, and in relation to the greater formation of “phage-lysosomes” observed in SHK-1 cells when they are incubated with PQSE, previous studies report that cholesterol accumulation in the phagosome membrane can alter its formation and maturation [99]. An excess of cholesterol in the early or late phagosome membrane prevents its maturation by affecting ATPase adhesion. Cholesterol accumulation has detrimental effects on phagosome maturation by preventing the activation of the small Rab ATPase Rab7, which is key in the phagosome–lysosome fusion, because it sequesters Rab7 and its effectors in cholesterol-rich multilamellar compartments [99]. One option is that the extracts of *Quillaja* purified in saponins (PQSE), by interacting with cholesterol from membranes, are disturbing it and favoring the maturation of the phagosome, which was demonstrated in this study.

### 4.5. SHK-1 Cells Pre-Incubated with PQSE Reduce the Expression of dotB (T4-BSS) and chaPs Protein of P. salmonis

Currently, there are no reports in the literature indicating that adding purified *Quillaja saponaria* extracts (PQSEs) to cell lines induces changes in the expression of virulence factors of pathogenic bacteria in their infectious cycle, and little information is available on fish pathogens such as *P. salmonis*. This constitutes a contribution of this study on the way to elucidating the mechanisms of action that strengthen the host response to counteract the infectious process of *P. salmonis*.

SHK-1 cells, when treated with PQSE at physiological doses as low as 0.5 µg/mL, reduce the intracellular proliferation of *P. salmonis* by mechanisms including a reduction in the expression of virulence factors, such as *dotB* from T4-BSS and *chaPs* protein, for up to 72 hpi.

### 4.6. SHK-1 Cells Pre-Incubated con PQSE Modulate the Expression of Anti-Inflammatory IL-10 and Proinflammatory IL-12 Gene Expression, Favoring a Proinflammatory Environment in the Early Phase of P. salmonis Infection (<72 h)

Our study showed that SHK-1 cells pre-incubated with purified *Quillaja saponaria* extract (PQSE) modulate the balance of proinflammatory (*IL-12*) and anti-inflammatory (*IL-10*) interleukins.

The balance of anti-inflammatory (*IL-10*) and proinflammatory (*IL-12*, *IF-g*, and *IL-6*) cytokines modulates the composition and the maturation of the phagosome (phagosome conversion) in its fusion with the lysosome during the bacterial infection process [24,27]. The phagosome maturation process is also influenced by a group of Rab GTPase proteins that are part of the vesicle formation and transport system and participate in membrane fusion. *IL-10* strongly decreases the expression of the Rab5 component of the early endosome. Several studies demonstrate that the generation of an anti-inflammatory environment in the macrophage, through the induction of *IL-10*, is a strategy followed by different intracellular bacteria. LPS induces the production of *IL-10* in human macrophages through a MAP-kinase-dependent pathway [43]. *P. salmonis* induces the production of *IL-10* to a level higher than that of *IL-12*, favoring an anti-inflammatory environment in the macrophage that houses it (an “inactive” macrophage) [20,41].

The adjuvant properties of *Quillaja* saponins have been known for decades [100,101,102,103,104,105]. Quillaja saponins interact with cell membranes, and this property explains several of its biological effects [106] since by destabilizing it and generating, among other things, pores, it changes the permeability of the membrane [91].

*Quillaja saponins* (QSs) are currently part of immune-stimulating complexes (ISCOMs) [107,108] and adjuvant systems such as AS01, which is used in two candidate vaccines that have recently shown efficacy in phase III trials: a vaccine against malaria, RTS, and a vaccine against Herpes Zoster, HZ/su. This AS01 adjuvant system has also shown efficacy in candidate vaccines against Tuberculosis and the Human Immunodeficiency Virus/AIDS, since QS potentiates both the cellular and humoral immune response to purified antigens [109]. A purified fraction of Quillaja saponins (QS-21) has been reported to induce the release of Interleukin 1 Beta (IL-1β) and Caspase-1 dependent IL-18 in antigen-presenting cells, such as macrophages and dendritic cells, when simultaneously stimulated with the adjuvant Monophosphoryl Lipid A (MPL), a TLR4 agonist. Furthermore, these reports suggest that the ASC-NLRP3 inflammasome is responsible for QS-21-induced IL-1β)/*IL-18* release [105]. A recent report investigated other immunomodulation pathways and reported that QSs directly activate monocyte-derived dendritic cells (moCDs) in a lysosome-dependent manner [110]. This study further demonstrated that QSs induce a proinflammatory transcriptional program. These findings coincide with our study, where a higher *IL-12* expression was quantified. Changes in the lysosome membrane can also induce phosphorylation of the kinase Syk. The Welsby et al. [110] study also observed phosphorylation of Syk after QS addition, finding that Syk is a key signaling molecule for the dendritic cell response of moDCs to QS addition. Syk kinase induction stimulates NFkB as well. In the present study, we demonstrated that the pre-incubation of SHK-1 cells with PQSE (96% saponins *w*/*w*) induced the production of both proinflammatory (*IL-12*) and anti-inflammatory (*IL-10*) interleukins, favoring a greater production of proinflammatory (*IL-12*) interleukins during the acute phase of the *P. salmonis* infection (≤72 hpi), which is key in the activation of macrophages and the control of intracellular pathogens.

## 5. Conclusions

Through the different in vitro tests performed, this study showed that purified *Quillaja saponaria* extracts (PQSEs), without presenting a direct antimicrobial action on *P. salmonis*, significantly reduce the internalization and intracellular proliferation of the bacteria.

According to the results of the study, PQSEs activate a protective response mechanism in the host cell. This mechanism alters the strategy and infective cycle of *P. salmonis* by reversing the anti-inflammatory condition of an “inactive” macrophage to the proinflammatory condition of an “active” macrophage and promoting the formation of “phago-lysosomes”.

The results also mean that strategies that do not focus their effect on attacking the aggressor agent, but on strengthening the natural response mechanisms of the host, “preparing” and/or “reactivating” macrophages, are very valuable as fundamental actors in the process of preventing and reversing the infectious process of *P. salmonis*.

A concrete consequence of the results of this study is the potential to reduce and optimize the use of antimicrobials in the control of SRS, supported by the use of PQSE, directed at the host and not the pathogen.

Another consequence of the results is that, by not directing the attack specifically against the pathogen, the PQSEs reduce the probability that bacteria will generate resistance against them, since the low levels of PQSE required to strengthen the host (0.5 µg/mL) are harmless to the bacteria.

Collectively, these findings provide new insights into the direct cellular effects of PQSE, encompassing: (1) lower expression of virulence protein factors from the T4-BSS (*dotB*) and the chaperone HSP60 (*chaPs*); (2) the promotion of a proinflammatory transcriptional program (more *IL-12* than *IL-10*) invested in an active infection by the bacteria; and (3) a significant reduction in the intracellular multiplication of *P. salmonis* due the enhancement of the phagosome–lysosome fusion in the host’s head kidney macrophages (SHK-1).

Finally, the evidence presented in this study allows us to affirm our hypothesis that purified *Quillaja saponaria* extracts (PQSEs) can alter the infective process of *Piscirickettsia salmonis*, reducing its internalization and intracellular proliferation.

### Ending Considerations

Considering the limitations of in vitro studies, but based on the immunological and non-antimicrobial results obtained in the study, along with the fact than PQSEs are approved for human and animal consumption, the goal is to develop a formulation that can be supplied to animals via their diet as a natural, non-antibiotic alternative for the prevention and control of SRS in the salmon industry.

## Figures and Tables

**Figure 1 antibiotics-10-00847-f001:**
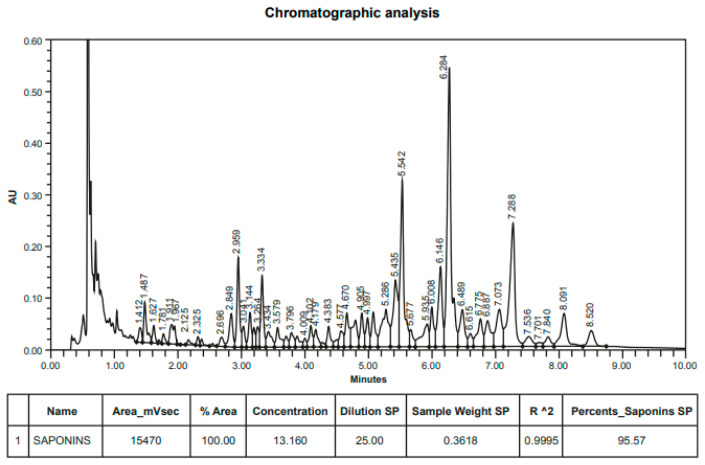
UHPLC of purified *Quillaja saponin extract* (PQSE) (96% saponins, *w*/*w*). UHPLC of PQSE from which almost all non-saponin fraction components (phenolics, polysaccharides) have been separated.

**Figure 2 antibiotics-10-00847-f002:**
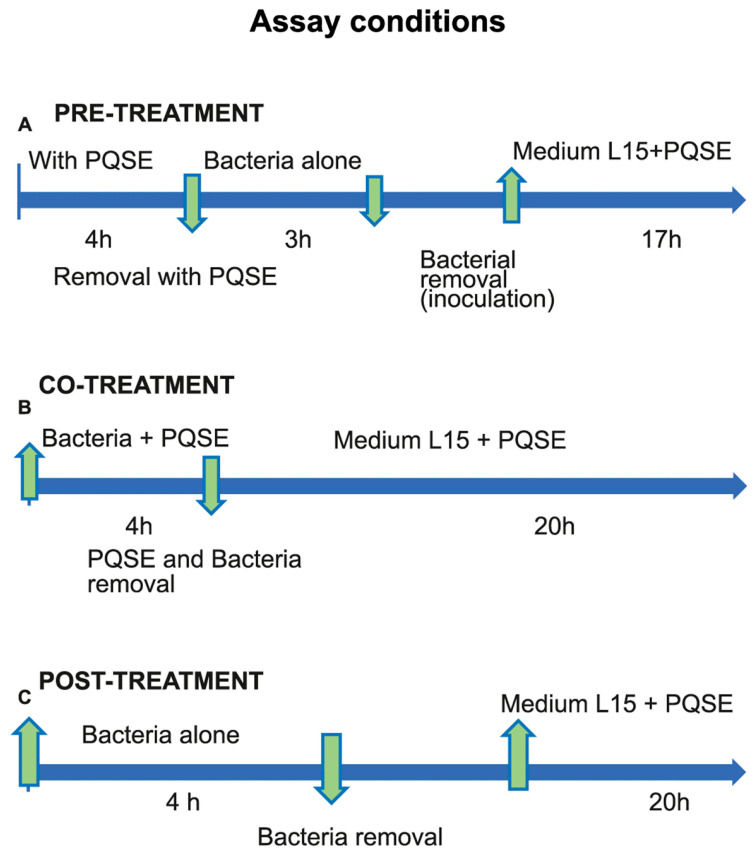
Assay conditions to phagosome-lysosome fusion measure. Three conditions, Pre-treatment, Co-treatment, and Post-treatment of SHK-1 cells with Quillaja Extract (PQSE), and challenged with *P. salmonis*, were used to evaluate the phagosome-lysosome fusion.

**Figure 3 antibiotics-10-00847-f003:**
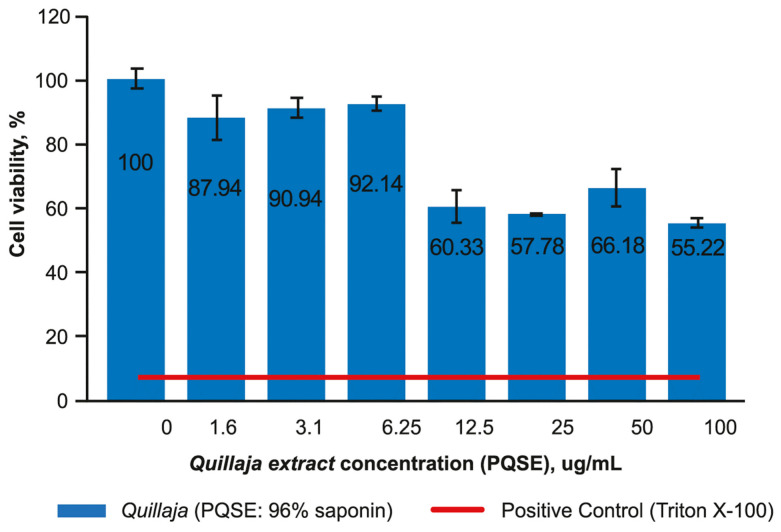
Viability of SHK-1 cells exposed to a High Purified Quillaja saponin Extract (PQSE). The image corresponds to Table 1. cell viability was higher than 90% until doses of 12.5 µg/mL of the extract (equivalent to 12.0 µg/mL of saponin). But at levels more than 12.5 ug/mL, the cell viability was reduced to 60%. The cell viability of the Positive Control (Triton X-100) was 39%. The viability was estimated as (Optical density of treated cells, OD)/(OD Control cells) × 100.

**Figure 4 antibiotics-10-00847-f004:**
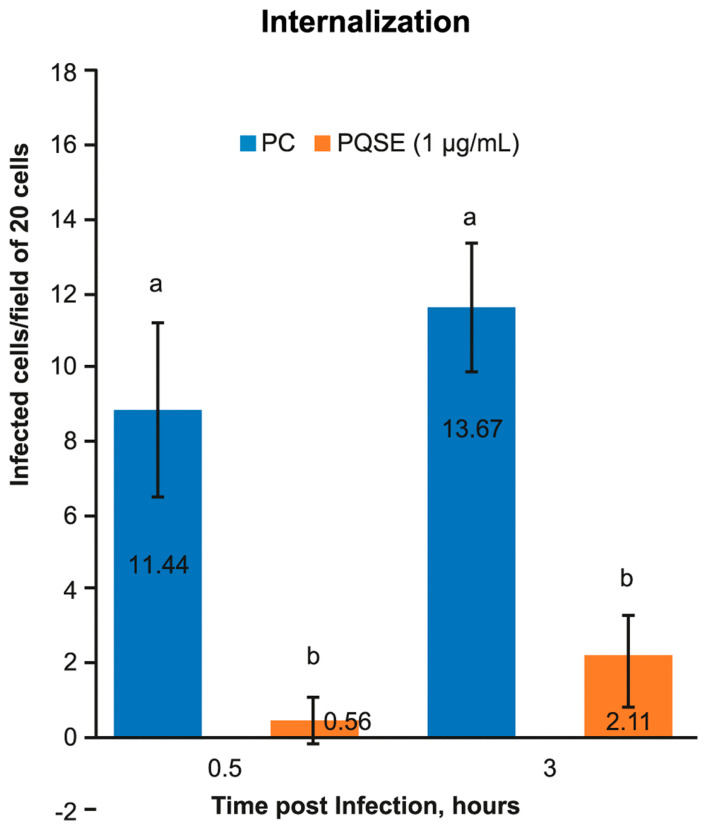
Internalization of *P. salmonis* in SHK-1 cells pre-incubated 4 h with 0.5 µg/mL of Quillaja saponaria Extract (PQSE). The internalization test for *P. salmonis* in SHK-1 cells, showed a significant reduction in the number of cells infected with the bacterium in the groups pre-incubated with PQSE, either, at 0.5 hpi (early internalization) as a 3 hpi (late internalization), with respect to the Positive Control (PC) (cells infected with *P. salmonis*). The test considered, two (2) treatments (PQSE & PC), three (3) repetitions/treatment, sixty (60) cells/repetition. The statistical analysis was performed using a Manova, to evaluate, time and treatments, independent variables. Different letters, a and b represent statistical differences at *p* < 0.05.

**Figure 5 antibiotics-10-00847-f005:**
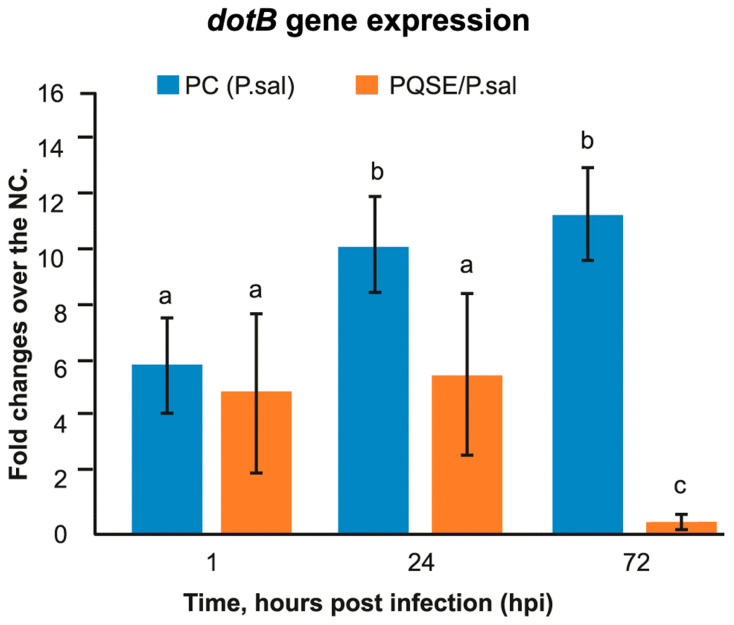
Gene expression analysis of *dotB* (T4-BSS) in SHK1 cells pre-incubated with 0.5 µg/mL of Quillaja extract (PQSE) and challenged with *P. salmonis*. *dotB* gene expression at 1, 2 and 72 hpi in cells pre-incubated with PQSE. Results are expressed as fold change relative to Negative Control (NC), as average values of triplicate determinations. SHK-1 cells pre-incubated with PQSE for 4 h, prior to a challenge with *P. salmonis*, significantly reduce (*p* < 0.05) the gene expression of the virulence factor *dotB*, compared to the Positive Control (PC: *P. salmonis*), at 24 and 72 hpi. NC: Negative Control (without PQSE and without *P. salmonis*); PC: Positive Control (*P. salmonis*) PQSE+*P.sal*: SHK-1 cells pre-incubated with PQSE and then challenged with *P. salmonis*. Different letters, a, b and c depict statistically significant differences according to the Mann-Whitney test (*p* <0.05).

**Figure 6 antibiotics-10-00847-f006:**
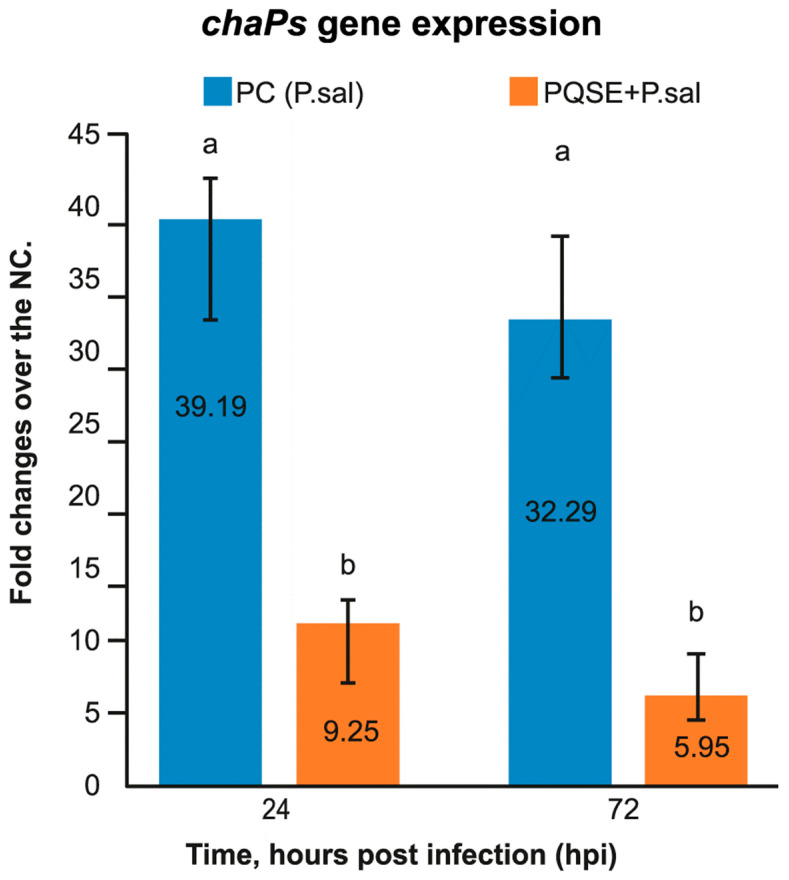
Gene expression analysis of *chaPs* at 24 and 72 hpi in SHK-1 cells pre-treated 4 h with 0.5 µg/mL of PQSE. Results are expressed as fold change relative to Negative Control (NC), as average values of triplicate determinations. SHK-1 cells pre-treated for 4 h with PQSE, prior to a challenge with *P. salmonis*, significantly reduced (*p* <0.05) the gene expression of *ChaPs*, compared to the PC, at 24 and 72 hpi. NC: Negative Control (without PQSE and without *P. salmonis*); PC: Positive Control (*P. salmonis*); PQSE/*P.sal*: SHK-1 cells pre-treated with PQSE and then challenged with *P. salmonis*. Different letters a and b depict statistically significant differences according to the Mann-Whitney test (*p* <0.05).

**Figure 7 antibiotics-10-00847-f007:**
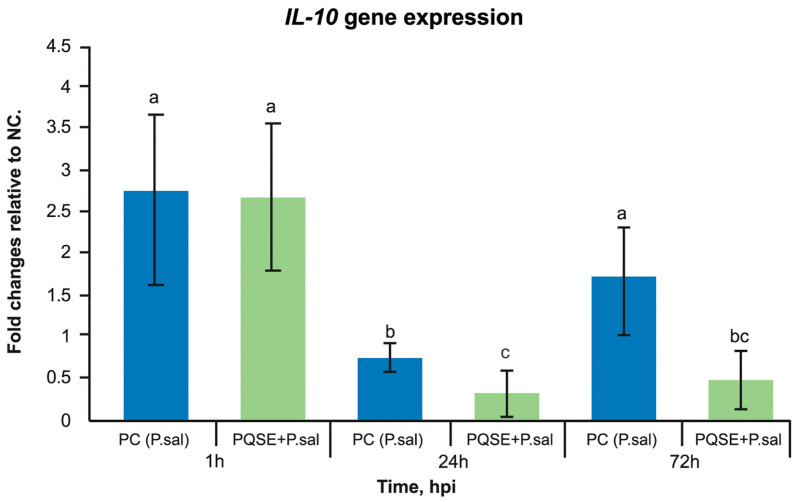
Gene expression analysis of IL-10 at 1, 24 and 72 hpi, in SHK-1 cells pre-treated with 0.5 µg/mL of PQSE. SHK-1 cells pre-incubated 4 h with PQSE, prior to a challenge with *P. salmonis*, significantly reduced (*p* < 0.05) the gene expression of Interleukin 10 (IL-10), with respect to NC and PC, at 24 and 72 hpi. The results are expressed as fold change relative to the NC, as average values of triplicate determinations. NC: Negative Control (without PQSE and without *P. salmonis*); PC: Positive Control (*P. salmonis*); PQSE/*P.sal*: SHK-1 cells pre-treated with PQSE and challenged with *P. salmonis*. Different letters, a, b and c depict statistically significant differences according to the Mann-Whitney test (*p* < 0.05).

**Figure 8 antibiotics-10-00847-f008:**
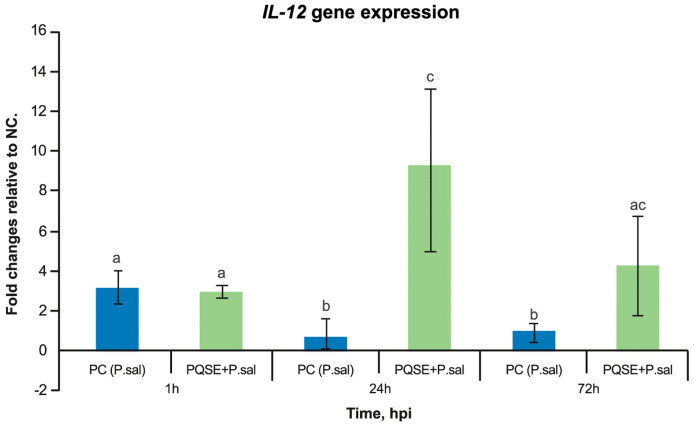
Gene expression analysis of IL-12 at 1, 24 and 72 hpi in SHK-1 cells pre-treated with 0.5 µg/mL of PQSE. SHK-1 cells pre-treated for 4 h with PQSE, prior to a challenge with *P. salmonis*, significantly increased (*p* < 0.05) the gene expression of IL-12, with respect to NC and PC, at 24 and 72 hpi. The results are expressed as fold change relative to the NC, as average values of triplicate determinations. NC: Negative Control (without PQSE and without *P. salmonis*); PC: Positive Control (*P. salmonis*); PQSE/*P.sal*: SHK-1 cells pre-treated with PQSE and challenged with *P. salmonis*. Different letters, a, b and c depict statistically significant differences according to the Mann-Whitney test (*p* < 0.05).

**Figure 9 antibiotics-10-00847-f009:**
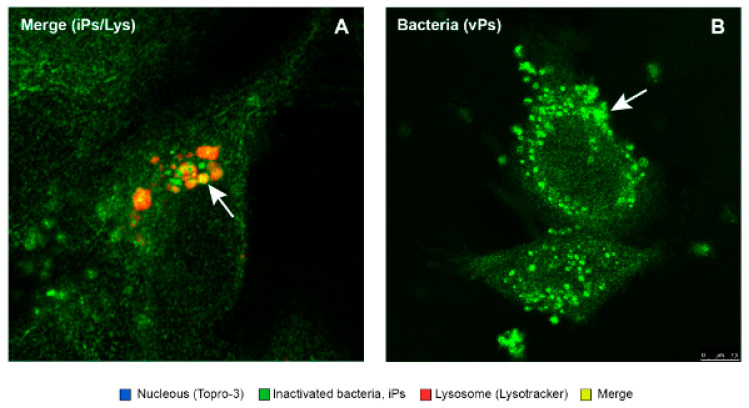
Colocalization of *P. salmonis* (virulent and inactivated) with lysosomes in SHk-1 cells. Intracellular localization of virulent *P. salmonis* LF89 (vPs) and formaldehyde-inactivated (iPs) strains within SHK-1 macrophages was evaluated by confocal microscopy. The confocal micrograph (panel (**A**)) shows inactivated *P. salmonis* (iPs) (in green) colocalized within lysosomes (red). The colocalization is visualized with a color between yellow and orange. This indicates the phagosome maturation of infected cells. (Panel (**B**)) shows a virulent strain of *P. salmonis* (vPs) (in green) that does not colocalize with lysosomes. VPs strains remain in their phagosome and do not mature to the P-L fusion, surviving and proliferating in macrophages.

**Figure 10 antibiotics-10-00847-f010:**
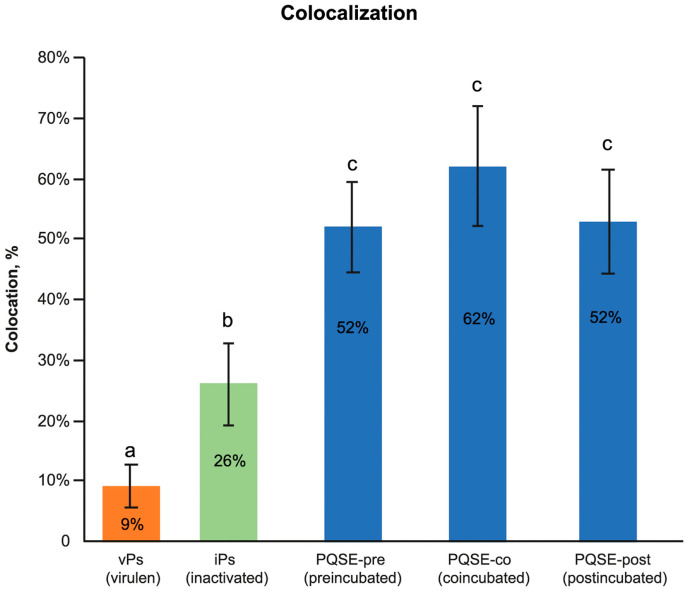
Colocalization of *P. salmonis* with lysosomes, in SHk-1 under Quillaja (PQSE). Intracellular localization of *P. salmonis* LF89 strains within SHK-1 macrophages was evaluated by confocal microscopy under three PQSE addition strategies. For the cell count, the design included 5 Treatments, 2 Rep/Treatment, 3 fields/Rep and 20 cells/field, for a total of 60 cells/Rep and 120 cells/treatment. It is observed that the treatments with the addition of 0.5 µg/mL of PQSE to the cells, independent of the time of its addition (Pre, Co, or post challenge with vPs), colocalized significantly more than the inactivated (iPs) and virulent (vPs) *P. salmonis*. No statistical differences (*p* > 0.05) were observed between the three PQSE addition strategies. The lowest colocalization was observed in cells challenged with the virulent strain (vPs), without PQSE. Different letters a, b and c depict statistically significant differences (*p* < 0.05).

**Figure 11 antibiotics-10-00847-f011:**
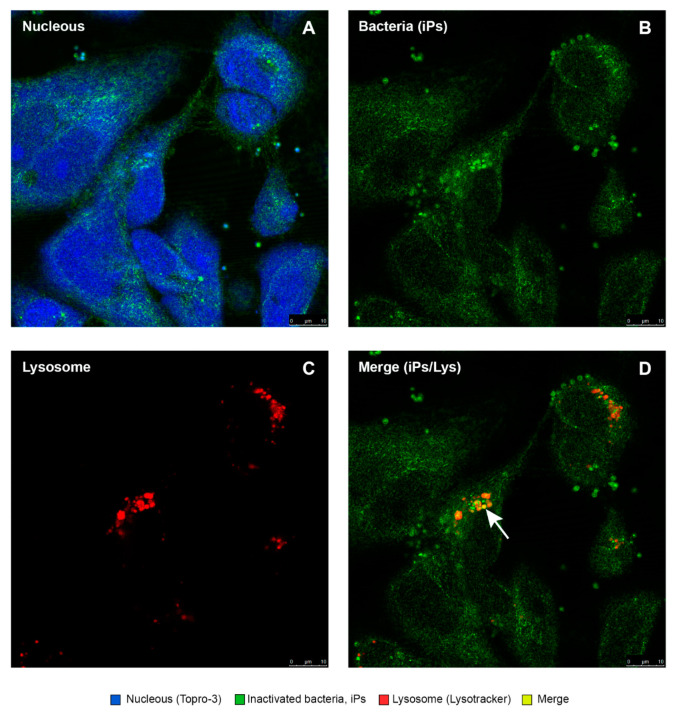
Intracellular colocalization of inactivated *P. salmonis* in SHK-1 cells. (**A**) The blue color shows the nuclei of the cells marked with TOPRO-3. (**B**) The green color shows inactivated bacteria (iPs). (**C**) Lysosomes were observed and are marked with Lysotracker Red DND-99. (**D**) The colocalization of the iPs within lysosomes is displayed in a yellow-orange color. This colocalization is a sign of phagosome–lysosome fusion.

**Figure 12 antibiotics-10-00847-f012:**
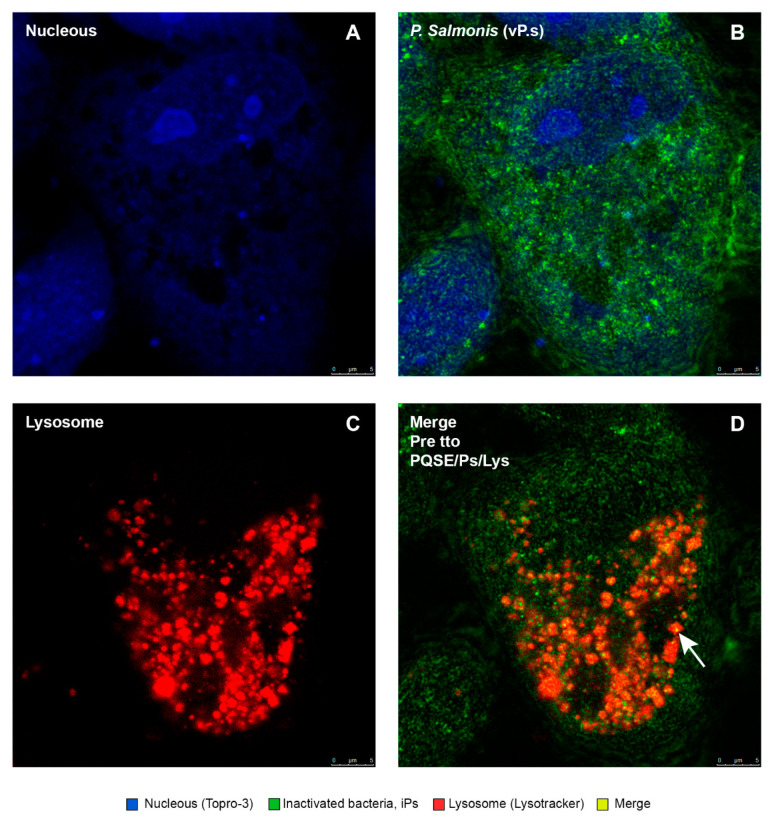
Location of *P. salmonis* in SHK-1 cells pre-treated with 0.5 µg/mL of PQSE (24 hpi). Photomicrographs show the location of the virulent *P. salmonis* (vPs) strain LF89 in SHK-1 cells pre-incubated with PQSE for 4 h, prior to challenge with the bacterium. (**A**) Nuclei of cells marked with TOPRO-3 (blue) can be observed. (**B**) Virulent bacteria (vPs) (green) can be observed. (**C**) Lysotracker Red DND-99-labeled lysosomes distributed throughout the cell (red) can be observed, and (**D**) vPs bacteria co-located within lysosomes were observed to be yellow-orange as an indicator that there was maturation of the phagosome that contained the bacterium and phagosome–lysosome fusion.

**Figure 13 antibiotics-10-00847-f013:**
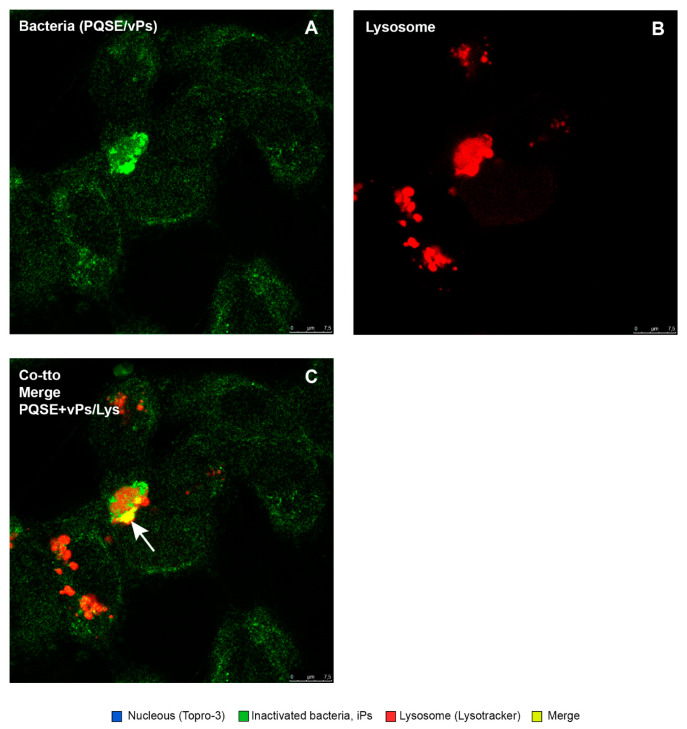
Location of *P. salmonis* in SHK-1 cells in a co-treatment with 0.5 µg/mL of PQSE (24 hpi)**.** Photomicrographs show the location of the virulent *P. salmonis* (vPs) strain LF89 in SHK-1 cells in a co-treatment with PQSE and simultaneous challenge with the bacterium (vPs) for 4 h. (**A**) The nuclei of the cells marked with TOPRO-3 (blue) can be observed. (**B**) Virulent bacteria (vPs) (green) can be observed. (**C**) Lysosomes marked with Lysotracker Red DND-99 (red) can be observed, and vPs bacteria located inside lysosomes were observed to be orange-yellow as an indicator that there was maturation of the phagosome that contained the bacterium and phagosome–lysosome fusion.

**Figure 14 antibiotics-10-00847-f014:**
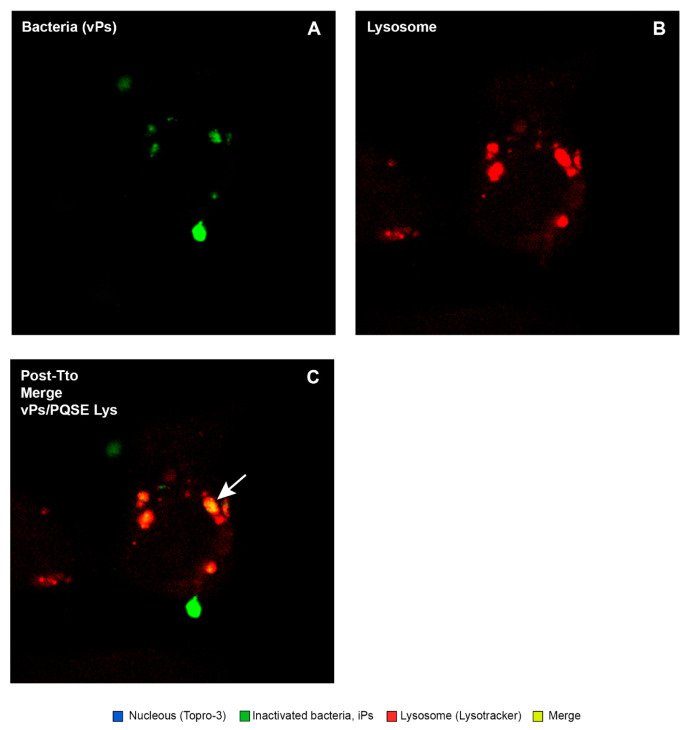
Location of *P. salmonis* in SHK-1 cells in post-treatment with 0.5 µg/mL of PQSE (24 hpi). The microphotographs show the location of the virulent *P. salmonis* (vPs) strain LF89 in SHK-1 cells treated with PQSE, 3 h after challenge with vPs. (**A**) The nuclei of the cells marked with TOPRO-3 (blue) can be observed. (**B**) Virulent bacteria (vPs) (green) can be observed. (**C**) The Lysotracker Red-labeled Lysosomes can be observed (red), and vPs bacteria located inside lysosomes were observed to be orange-yellow as an indicator that there was maturation of the phagosome that contained the bacterium and phagosome–lysosome fusion.

**Table 1 antibiotics-10-00847-t001:** Primers for the real-time PCR gene expression analysis used in our study.

Gene	Primer Sequence	Reference
*sdhA*	For: 5′-ATTTCTTTGGAGCTACGTGAAG-3′	Flores-Herrera et al., 2018
	Rev: 5′-CCACCCATCATATAATGACAAG-3′	
*elf1A*	For: 5′-GTC TAC AAA ATC GGC GGT AT-3′	Peña et al., 2010
	Rev: 5′-CTT GAC GGA CAC GTT CTT GA-3′	
*ITS*	RTS1; For-223: 5′-TGATTTTATTGTTTAGTGAGAATGA-3′	Marshall et al., 1998
	RTS4; Rev-459: 5′-ATGCACTTATTCACTTGATCATA-3′	
*dotB*	Ps-dotB-For: 5′-GCT ACA TCT CCA TTT CTT GAC CAT TTC-3′	Gómez et al., 2013
	Ps-dotB-Rev: 5′- GCA TTA GTG CCG AGC ATT ACA GG-3′	
*chaPs*	For: 5′-GATGAAAGAGAAGAAAGACCGC-3′	Marshall et al., 2007
	Rev: 5′-ATGGGCGGCATGGGCGGCATGATG-3′	
*IL-10*	For: GCCCTTCAGTAACTTACACAGATGGAC	Harun et al., 2011
	Rev: GTCGTTGTTGTTCTGTGTTCTGTTGT	
*IL-12*	For: CCCAACACGGACAGGAACAC	Wang et al., 2014
	Rev: GCCCTTCAGTAACTTACACAGATGGAC	

**Table 2 antibiotics-10-00847-t002:** Absolute quantification of *P. salmonis* DNA at 1 hpi and 72 hpi in SHK-1 cells. The pre-treatment of SHK-1 cells with PQSE (0.5 µg/mL) for 4 h significantly (*p* < 0.05) reduced the intracellular proliferation of *P. salmonis* (by 80% at 1 hpi (Panel A) and 76% at 72 hpi) as measured by absolute quantification of bacterial DNA.

**Panel A—Intracellular Proliferation of *P. salmonis* at 1 hpi**				
**Treatment**	**Time** **hpi**	**Ct** **(dRn)**	**No. Copies/Samples**	**%** **Inhibition**
Negative Control, NC	1	38.51 ± 0.11	0.0	N/A
Positive Control, PC (*P. salmonis*)	1	29.68 ± 0.12	2377 ± 1642	0
PQSE (0.5 µg/mL) + *P. salmonis*	1	31.66 ± 0.07	467 ± 304	80
**Panel B—Intracellular Proliferation of *P. salmonis* at 72 hpi**				
**Treatment**	**Time** **hpi**	**Ct (dRn)**	**Number of Copies/Samples**	**%** **Inhibition**
Negative Control, NC	72	38.41 ± 0.22	0.0	
Positive Control, PC (*P. salmonis*)	72	24.44 ± 0.01	33,479 ± 7730	
PQSE (0.5 µg/mL) + *P. salmonis*	72	27.58 ± 1.18	8026 ± 4514	76%

## Data Availability

The data from our research are available in this article. Figures and Tables are in a printable (PDF) and editable (AI) version. The files for printing the cell figures (Figure 10 and Figure 12, Figure 13, and Figure 14) are those that are in the TIFF format. Additionally, the end file, “cell figures.PSD”, contains all the figures with cell photographs and is editable (original). All images were exported at a size of 300 dpi: https://www.dropbox.com/sh/hjrmuzbp0aucsjt/AADsyNKW98I9nsgoVb7Q-RuSa?dl=0 (accessed on 23 March 2021).

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
