# Peer review of "The Phagosome–Lysosome Fusion Is the Target of a Purified Quillaja saponin Extract (PQSE) in Reducing Infection of Fish Macrophages by the Bacterial Pathogen Piscirickettsia salmonis"

_antibiotics, 2021, doi:10.3390/antibiotics10070847_

Round 1

Reviewer 1 Report

This is very interesitng story, but it needs more work and extensive editorial work.

Authors should add data obtained with real positive control drugs in all experiments.

A lot of typo and grammatical errors should be revised throughout the manuscript....italic for microorganism names...High Purified Quillaja saponin Extracts (PQSE) should be Purified Quillaja saponin Extract (PQSE) etc.

Abstract is really bad. Please rewrite it.

Please mention the concentration of PQSE in legends of Fig

Authors have mentioned as below...Authors should include additional data (as mechanism study) to expain how it can show below

1) less expression of 808 virulence protein factors from the T4-BSS (dotB) and the chaperone HSP60 (chaPs),

2) 809 promotion of a pro-inflammatory transcriptional program (more IL-12 than IL-10) 810 invested in an active infection by the bacteria and,

3) significant reduction of 811 intracellular multiplication of P.salmonis due the enhancement of the Phagosome-812 Lysosome fusion in the host head kidney macrophages (SHK-1).

Author Response

Dear Reviwer 1. Hello.

Thanks a lot for the suggestions. All were considered.

Please see the attached word file with the answers, and some papers for support it.

Best regards.

Hernán

Reply to the Review Report (Reviewer 1)

Authors should add data obtained with real positive control drugs in all experiments.

Since ours MIC studies and others (Cañon-Jones et al, 2020) have shown that PQSE does not exert a direct anti-microbial effect on P. salmonis, the study of the putative mechanisms of action were focused on the host cell. Therefore, we did not consider including drugs against the bacteria as a positive control in the experiments.

Based on a minimum inhibitory concentration (MIC) test of PQSE in cell-free culture medium (BM3), our study showed that PQSE per se, does not affect the viability of P. salmonis at cell physiological doses (< 12.0 µg/mL). MIC value determination demonstrated that to reduce P. salmonis viability PQSE should be used at doses greater than 30,000 µg/mL. On the other hand, efficacy studies of PQSE on P.salmonis in the macrophage-like cell line SHK-1 showed that only 0.5 µg/mL of PQSE was required to:  a) modulate cytokine balance (IL-12/IL-10); b) reduce the expression of pathogen virulence factors dotB and chaPs, and c) induce phagosome-lysosome binding, all of which resulted in reduced invasion and multiplication of P.salmonis within the macrophage.

A lot of typo and grammatical errors should be revised throughout the manuscript....italic for microorganism names...High Purified Quillaja saponin Extracts (PQSE) should be Purified Quillaja saponin Extract (PQSE) etc.

Done. This observation was accepted, and the changes were made in the new version of the paper.

Abstract is really bad. Please rewrite it.

Done. The abstract was rewrite in the new version of the paper.

Please mention the concentration of PQSE in legends of Fig

Done. This observation was accepted, and the concentration of PQSE was included in the corresponding figures in the new version of the paper.

Authors should include additional data (as mechanism study) to explain how it can show below.

  • Less expression of virulence protein factors from the T4-BSS (dotB) and the chaperone HSP60 (chaPs),

The Dot/Icm T4-BSS genes expression are upregulated during the different stages of infection process of P. salmonis, specially during the early phases post-infection. Considering the down regulation of the pathogenic genes dotB and chaPs in PQSE treated cells vs the control cells, we speculate that PQSE can affect the bacterial infection process, reducing the expression of the pathogenic genes, however, validation experiments are needed to elucidate this possibility.

On the other hand, a valid question could be posed. There was a relationship between the lower expression of P. salmonis dotB and chaPs pathogenic genes with the lower number of bacteria observed in PQSE-treated cells, or not?

When we developed the gene expression kinetics, either for dotB, chaPs, IL-10, IL-12, and the determination of the bacterial load (bacterial DNA) by qRT-PCR, we performed a relative quantification, "normalizing" with respect to the Control treatments, so that all the samples evaluated could be comparable in gene expression, using the calculation of the Ct value (threshold cycle). This "normalization" allowed us to compare gene expression, independent of the number of bacteria in each sample. Therefore, we consider that the lower expression of the genes in this study (dotB, chaPs, IL-10, IL-12) did not depend on the decrease in the number of bacteria in the PSQE-treated sample.

  • Promotion of a pro-inflammatory transcriptional program (more IL-12 than IL-10) invested in an active infection by the bacteria.

Previous reports (Welsby et al 2017), have shown that Quillaja saponins can induce direct activation of antigen presenting cells (macrophages, dendritic cells) as well as expression of cytokines such as IL-12, even in the absence of pathogens. This direct activation is due to the ability of saponins to integrate into the macrophage through cholesterol-dependent endocytosis, and as such, accumulate in the membrane of the endosome and lysosome, destabilizing the membrane of these organelles, and subsequently activating Syk kinases, key signaling molecules in the activation of  NFkB and the induction of a proinflammatory transcriptional program. In our understanding, this evidence shows that Quillaja saponins could directly modulate cytokine expression in cells irrespective of a, pathogen infection.

Other report in a murine model (Katayama et al, 2006) showed that the oral supplementation of Quillaja saponin can change the profile of cytokines associated with allergy and the antigen-specific immune response through regulation of Th1/Th2 cytokines balance. The study found that Quillaja saponaria Extracts promote a Th1-dominant immune response (proinflammatory) and can suppress the ovalbumin-induced IgE-mediated allergic response. That study is other in vivo evidence that Quillaja saponin modulate the balance of cytokines, even without a pathogen infection.

  • Significant reduction of intracellular multiplication of salmonis due the enhancement of the Phagosome-Lysosome fusion in the host head kidney macrophages (SHK-1).

According to our study, cells treated with PQSE favored the phagosome-lysosome fusion. With relation to the greater formation of "Phage-Lysosomes" observed in SHK-1 cells treated with PQSE, previous studies from Barry (2011) and Thi (2012) demonstrated than the balance of anti-inflammatory (IL-10) and pro-inflammatory (IL-12) cytokines may modulates the composition and the “maturation of the phagosome” (phagosome conversion) and its fusion with the lysosome, during the bacterial infection process. The phagosome maturation process is also influenced by a group of Rab GTPases proteins, which are part of the vesicle formation and transport system and membrane fusion. IL-10 strongly decreases the expression of Rab5 component of the early endosome.

On the other hand, Huynh (2008) reported that cholesterol accumulation in the phagosome membrane can alter its formation and maturation. An excess of cholesterol in the early or late phagosome membrane prevents its maturation by affecting ATPase adhesion. Cholesterol accumulation has detrimental effects on phagosome maturation by preventing the activation of the small Rab ATPase Rab7, key in the phagosome-lysosome fusion, because it sequesters Rab7 and its effectors in cholesterol-rich multilamellar compartments. Quillaja saponins (PQSE) has the ability of form complex with cholesterol at cell membranes disturbing it and increasing its plasticity. This effect can favor the maturation of the phagosome and promoting the formation of phago-lysosomes. According to that, we speculate that the higher phagolysosome formation we found in cells treated with PQSE, could be explained in part by this interaction of Quillaja saponins with cholesterol, however new validation experimental research is needed to demonstrate this mechanism.

For example, Arayan et al (2015) found that RAW 264.7 cells incubated with RGSF-A saponins from the red ginseng and infected with Brucella abortus induced an immune modulation which was manifested with the inhibition of bacterial uptake and intracellular replication of the bacteria in this king of macrophages. This study demonstrated a downregulation of MAPKs and inhibition of bacterial penetration by restricting F-actin polymerization. The study suggested that the reduction of invasion of B. abortus in RGSF-A treated cells may be due to the downregulation of MAPKs (ERK1/2 and p38α) kinases.

PQSE can induce interferon gamma (INF-gamma), which can promote the phagosome- lysosome fusion.

 PQSE given into the diet of salmon fish as a non-pharmacologic alternative to antibiotics in the control of Piscirickettsiosis (SRS) are being developed at industrial conditions in this moment in Chile (more than 1 million salmon fish/trial) with salmon companies like MOWI. This industrial validation trials, include the monitor of mortality, the use of antibiotic in case of SRS outbreak, and the evaluation of the immune status of Atlantic salmon (Salmo salar) post smolt and during the fattening period. Considering that PQSE do not have a direct antimicrobial effect on P. salmonis, but can modulate the immune response of the host, studies are evaluating the kinetics of innate and the adaptive immune response markers, like the IL-12/IL-10 balance, the CD4+ and CD8+ T-cell gene expression, the induction of Interferon gamma (INF-gamma), antimicrobial peptides AMP (Cathelicidine), and the production of IgT and IgM immunoglobulins at kidney, gut and gills level.  

Until now, the cited experiences at Mowi and other salmon fish companies have demonstrated at commercial scale conditions, that PQSE given into the diet of salmon fish at least one month before the natural challenges of P. salmonis, can modulate the innate and adaptive immune response in salmon fish, including an adequate balance of IL-12/IL-10),  induction of both, the cellular (CD8+ and CD4+ T Cells) and the humoral arms of the adaptive immune response, including the induction of gamma interferon (Type II Interferon).

Although in our in vitro study, the quantification of INF-gamma was not performed because the macrophage cell line used (SHK-1) is not suitable for quantifying this kind of protein, the cited in vivo studies at industrial scale conditions have demonstrate that PQSE induces INF-gamma, which is involved in the activation of macrophages, favoring phagosome-lysosome binding and its killing capacity against intracellular pathogens such as P. salmonis (Alvarez et al, 2016). This could be another mechanism by which PQSE could induce the phagosome-lysosome fusion in in vivo conditions.

Finally, we would like to express that Patents using Purified Quillaja saponaria Extracts (PQSE) in the control of Piscirickettsia salmonis in salmon fish, have recently been granted in Norway and United States, and

  • Norwegian Industrial Property Office, Patent N° 345485: Use of Quillaja saponaria extracts for the prevention and control of bacterial infections in fish and compositions comprising such extracts.
  • US Patent N°, 10.987,393 B2 (April 17, 2021): Method for preventing and controlling bacterial infections in salmonid fish using Quillaja saponaria

Reviewer 2 Report

The work deals with the use of Purified Quillaja saponin extracts (PQSE) to reduce the infectious process of P. salmonis.It was demonstrated that PQSE reduced the invasion and intracellular proliferation of P. salmonis in a macrophage fish cell line of SHK-1.

The manuscript is well structured and supported with related figures and experimental results.  

It can be published, after some minor corrections listed below:

C1) Please make sure that all P. salmonis and every other need to be written in italic in the text.

C2) Line 43 : The format of the references needs to be written in numbers as all others and numbered as 8 to 11.

C3) Some of the reference numbers are needed to be combined. Which are on the Line 42, 124,128,492,560,572,589,676 and 754.

C4) Line 19 :  the abbreviation of PQSE was given on Line 19 for Purified Quillaja saponin extracts and on the  line 546 for  High Purified Quillaja saponaria Extracts. Please set PQSE for one of those.

C5) Line 266 : The table needs to be numbered as Table 1 and the rest of the tables should re-numbered accordingly.

C6) Typos of the double dot needs to be corrected on the line 358 and 563.

C7) Line 632: An study supposed to be A study, please check it.

Author Response

Dear Reviwer 2. Hello.

Thanks a lot for the suggestions. All were considered.

Please see the attachment file in the box.

Best regards.

Hernan 

Round 2

Reviewer 1 Report

Authors have fully addressed and therefore it is now acceptable.